# zPROBE:
# Zero Peek Robustness Checks for Federated Learning

**Zahra Ghodsi**[1*]**, Mojan Javaheripi**[1*]**, Nojan Sheybani**[1*]**, Xinqiao Zhang**[1*]**,**
**Ke Huang**[2]**, Farinaz Koushanfar**[1]
[1]University of California San Diego, [2]San Diego State University
[1]{zghodsi, mojan, nsheyban, x5zhang, farinaz}@ucsd.edu
[2]khuang@sdsu.edu

## Abstract

Privacy-preserving federated learning allows multiple users to jointly train a model with coordination of a central server. The server only learns the final aggregation result and not the individual user updates. However, keeping the individual updates private allows malicious users to perform *Byzantine attacks* and degrade the accuracy without being detected. Best existing defenses against Byzantine workers rely on robust rank-based statistics, e.g., median, to find malicious updates. However, implementing privacy-preserving rank-based statistics is nontrivial and not scalable in the secure domain, as it requires sorting all individual updates. We establish the first private robustness check that uses high break point rank-based statistics on aggregated model updates. By exploiting randomized clustering, we significantly improve the scalability of our defense without compromising privacy. We leverage our statistical bounds in zero-knowledge proofs to detect and remove malicious updates without revealing the private user updates. Our novel framework, zPROBE, enables Byzantine resilient and secure federated learning. Empirical evaluations demonstrate that zPROBE provides a low overhead solution to defend against state-of-the-art Byzantine attacks while preserving privacy.

## 1   Introduction

Federated learning (FL) has emerged as a popular paradigm for training a central model on a dataset distributed amongst many parties, by sending model updates without requiring the parties to share their data. However, model updates in FL can be exploited by adversaries to infer properties of the users' private training data [1]. This lack of privacy prohibits the use of FL in many machine learning applications that involve sensitive data such as healthcare information [2, 3] or financial transactions [4]. As such, existing FL schemes are augmented with privacy-preserving guarantees. Recently proposed secure aggregation protocols use cryptography [5, 6, 7] to ensure that the server does not learn individual user updates, but only a final aggregate with contribution from several users. Hiding individual updates from the server opens a large attack surface for malicious clients to send invalid updates that compromise the integrity of distributed training.

*Byzantine attacks* on FL are carried out by malicious clients who manipulate their local updates to degrade the model performance [8, 9, 10]. Popular high-fidelity Byzantine-robust aggregation rules rely on rank-based statistics, e.g., trimmed mean [11, 12], median [11], mean around median [13, 14], and geometric median [15, 16, 13]. These schemes require sorting of the individual model updates across users. As such, using them in secure FL is nontrivial and unscalable to large number of users since the central server cannot access the (plaintext) value of user updates.

In this work we address aforementioned challenges and provide high break point Byzantine tolerance using rank-based statistics while preserving privacy. We propose a median-based robustness check that derives a threshold for acceptable model updates using securely computed mean over random user clusters. Our thresholds are dynamic and automatically change based on gradient distributions. Notably, we do not need access to individual user updates or public datasets for our defense. We leverage the computed thresholds to identify

---

*Equal contribution

2022 Trustworthy and Socially Responsible Machine Learning (TSRML 2022) co-located with NeurIPS 2022.

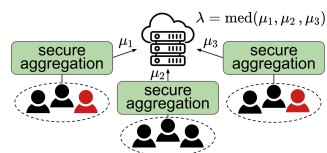
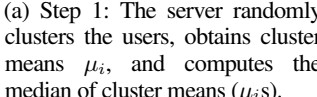
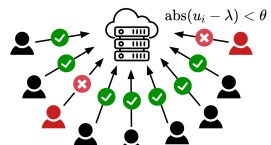
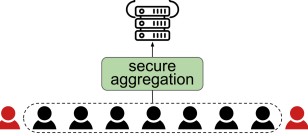

(a) Step 1: The server randomly clusters the users, obtains cluster means $\mu_i$, and computes the median of cluster means ($\mu_i$s).

(b) Step 2: Each client provides a ZKP attesting that their update is within the threshold from the median of cluster means.

(c) Step 3: Clients marked as benign participate in a final round of secure aggregation and the server obtains the result.

Figure 1: High level description of zPROBE robust and private aggregation.

and filter malicious users in a privacy-preserving manner. Our Byzantine-robust framework, zPROBE, incorporates carefully crafted zero-knowledge proofs [17, 18] to check user behavior and identify possible malicious actions, including sending Byzantine updates or deviating from the secure aggregation protocol. As such, zPROBE guarantees correct and consistent behavior in the challenging malicious threat model.

We incorporate probabilistic optimizations in the design of zPROBE to minimize the overhead of our zero-knowledge checks, without compromising security. By co-designing the robustness defense and cryptographic components of zPROBE, we are able to provide a scalable and low overhead solution for private and robust FL. Our construction is the first of its kind with cost that grows sub-linearly with respect to the number of clients. zPROBE performs an aggregation round on ResNet20 over CIFAR-10 with only sub-second client compute time. In summary, our contributions are:

- Developing a novel privacy-preserving robustness check based on rank-based statistics. zPROBE is robust against various Byzantine attacks with $0.5-2.8\%$ higher accuracy compared to prior work on private and robust FL.

- Enabling private and robust aggregation in the malicious threat model by incorporating zero-knowledge proofs. Our Byzantine-robust secure aggregation, for the first time, scales sub-linearly with respect to number of clients.

- Leveraging probabilistic optimizations to reduce zPROBE overhead without compromising security, resulting in orders of magnitude client runtime reduction.

## 2 Cryptographic Primitives

**Shamir Secret Sharing** [19] is a method to distribute a secret $s$ between $n$ parties such that any $t$ shares can be used to reconstruct $s$, but any set of $t-1$ or fewer shares reveal no information about the secret.

**Zero-Knowledge Proof (ZKP)** is a cryptographic primitive between two parties, a prover $\mathcal{P}$ and a verifier $\mathcal{V}$, which allows $\mathcal{P}$ to convince $\mathcal{V}$ that a computation on $\mathcal{P}$'s private inputs is correct without revealing the inputs. We use the Wolverine protocol [17] with highly efficient $\mathcal{P}$ in terms of runtime, memory usage, and communication. Details of the Wolverine protocol are included in Appendix A.

**Secure FL Aggregation** includes a server and $n$ clients each holding a private vector of model updates with $l$ parameters $\boldsymbol{u} \in \mathbb{R}^l$. The server wishes to obtain the aggregate $\sum_{i=1}^{n} \boldsymbol{u_i}$ without learning any of the individual client updates. [5] and follow up [6] propose a secure aggregation protocol using low-overhead cryptographic primitives such as one-time pads. Each pair of clients $(i,j)$ agree on a random vector $\boldsymbol{m_{i,j}}$. User $i$ adds $\boldsymbol{m_{i,j}}$ to their input, and user $j$ subtracts it from their input so the masks cancel out when aggregated. To ensure privacy in case of dropout or network delays, each user adds an additional random mask $\boldsymbol{r_i}$. Users then create $t$-out-of-$n$ Shamir shares of their masks and share them with other clients. User $i$ computes their masked input as follows:

$$\boldsymbol{v_i} = \boldsymbol{u_i} + \boldsymbol{r_i} - \sum_{0<j<i} \boldsymbol{m_{i,j}} + \sum_{i<j\leq n} \boldsymbol{m_{i,j}} \qquad (1)$$

Once the server receives all masked inputs, it asks for shares of pairwise masks for dropped users and shares of individual masks for surviving users (but never both) to reconstruct the aggregate value. The construction in [6] builds over [5] and improves the client runtime complexity to logarithmic scale rather than linear with respect to the number of clients. Note that the secure aggregation of [6, 5] assumes the clients are semi-honest and do not deviate from the protocol. However, these assumptions are not suitable for our threat model which involves malicious clients. We propose an aggregation protocol that benefits from speedups in [6] and is augmented with zero-knowledge proofs for the challenging malicious setting as described below.

| Algorithm 1: zPROBE secure aggregation | Algorithm 2: zPROBE correctness check |
|---|---|

**Algorithm 1: zPROBE secure aggregation**

**Input:** Shamir threshold value $t$, clients set $U$
**Round 1: Mask Generation** (*client* $i$)
    Generate key pair $(sk_i, pk_i)$, sample $b_i$
    $a_{i,j} \leftarrow \text{KeyAgreement}(sk_i, pk_j)$
    $\boldsymbol{m_{i,j}} \leftarrow \text{PRG}(a_{i,j})$, $\boldsymbol{r_i} \leftarrow \text{PRG}(b_i)$
    $\{s_j^{sk}\}_{j \in U}, \leftarrow \text{SS}(sk_i, t), \{s_j^b\}_{j \in U} \leftarrow \text{SS}(b_i, t)$
    Send $s_j^{sk}, s_j^b$ to client $j$
**Round 2: Update Masking**
*client* $i$:
    $\boldsymbol{v_i} \leftarrow \boldsymbol{u_i} + \boldsymbol{r_i} - \sum\limits_{0<j<i} \boldsymbol{m_{i,j}} + \sum\limits_{i<j\leq U} \boldsymbol{m_{i,j}}$
    Authenticate $\boldsymbol{u_i}$ and send $\boldsymbol{v_i}$ to server
    Perform correctness check in Alg 2
*server:*
    Sample $q$ indices $S_i$ (Sec. 3.3) for client $i$
    Perform correctness check in Alg 2
**Round 3: Aggregate Unmasking** (*server*)
    $U_d \leftarrow$ dropped clients, $U_s \leftarrow$ surviving clients
    Collect $t$ shares of $\{s_i^{sk}\}_{i \in U_d}$ and $\{s_i^b\}_{i \in U_s}$
    $\text{Agg} \leftarrow \sum\limits_{i \in U_s} \boldsymbol{v_i} - \sum\limits_{i \in U_s} \boldsymbol{r_i} + \sum\limits_{i \in U_s, j \in U_d} \boldsymbol{m_{i,j}}$

**Algorithm 2: zPROBE correctness check**

**Client input:** $b_i$, $a_{i,j}$, authenticated $\boldsymbol{u_i}$
**Public input:** $\boldsymbol{v_i}$, indices set $S_i$, clients set $U$
**Circuit:**
    $check = 1$
    **for** $k$ in $S_i$
        $\hat{r}_i^k \leftarrow \text{PRG}^k(b_i)$
        **for** $j$ in $U$
            $\hat{m}_{i,j}^k \leftarrow \text{PRG}^k(a_{i,j})$
        $\hat{v}_i^k \leftarrow u_i^k + \hat{r}_i^k - \sum\limits_{0<j<i} \hat{m}_{i,j}^k + \sum\limits_{i<j\leq n} \hat{m}_{i,j}^k$
        $check = check \wedge (\hat{v}_i^k = v_i^k)$
    **return** $check$

**Algorithm 3: zPROBE robustness check**

**Client input:** Authenticated $\boldsymbol{u_i}$
**Public input:** $\boldsymbol{\lambda}, \boldsymbol{\theta}$, indices set $S_i$
**Circuit:**
    $check = 1$
    **for** $k$ in $S_i$
        $check = check \wedge (|u_i^k - \lambda^k| < \theta^k)$
    **return** $check$

## 3 Methodology

**Threat Model.** We aim to protect the privacy of individual client updates as they leak information about clients' private training data. No party should learn any information about a client's update other than the contribution to an aggregate value with inputs from a large number of other clients. We also aim to protect the central model against Byzantine attacks, i.e., when a malicious client sends invalid updates to degrade the model performance. We consider a semi-honest server that follows the protocol but may try to learn more information from the received data. We assume a portion of clients are malicious, i.e., arbitrarily deviating from the protocol, or sending erroneous updates to cause divergence in the central model. Notably, we assume the clients may: ① perform Byzantine attacks by changing the value of their model update to degrade central model performance, ② use inconsistent update values in different steps of the secure aggregation protocol, ③ perform the masked update computation in Eq. 1 incorrectly or with wrong values, and ④ use incorrect seed values in generating masks and shares. To the best of our knowledge, zPROBE is the first single-server framework with malicious clients that is resilient against such an extensive attack surface, supports client dropouts, and does not require a public clean dataset.

**zPROBE Overview.** zPROBE comprises two main components, namely, secure aggregation, and robustness establishment. We propose a new secure aggregation protocol for malicious clients in Sec. 3.1. Our proposed method to establish robustness is detailed in Sec. 3.2. We design an adaptive Byzantine defense that finds the dynamic range of acceptable model updates per iteration. Using the derived bounds, we perform a secure range check on client updates to filter Byzantine attackers. Our robustness check is privacy-preserving and highly scalable. The proposed robust and private aggregation is performed in three steps as illustrated in Fig. 1. First the server clusters the clients randomly into $c$ clusters. Each cluster $c_j$ then performs zPROBE's secure aggregation protocol. The server obtains the aggregate value $\boldsymbol{\alpha_j}$ and the mean $\boldsymbol{\mu_j} = \boldsymbol{\alpha_j} / |c_j|$ for each cluster in plaintext. In the second step, the server uses the median $\boldsymbol{\lambda}$ of all cluster means to compute a threshold $\boldsymbol{\theta}$ for model updates. The values of median $\boldsymbol{\lambda}$ and threshold $\boldsymbol{\theta}$ are public, and broadcasted by the server to all clients. Each client $i$ then provides a zero-knowledge proof attesting that their update is within the threshold from the median, i.e., $\text{abs}(\boldsymbol{u_i} - \boldsymbol{\lambda}) < \boldsymbol{\theta}$. This ensures that clients are not performing Byzantine attacks on the central model (item ① in threat model). Users that fail to provide the proof are considered malicious and treated as dropped. The remaining users participate in a round of zPROBE secure aggregation and the server obtains the final aggregate result.

### 3.1 zPROBE Secure Aggregation

Alg. 1 shows the detailed steps for zPROBE's secure aggregation for $n$ clients consisting of three rounds. In round 1, each client $i$ generates a key pair $(sk_i, pk_i)$, samples a random seed $b_i$, and performs a key agreement protocol [20] with client $j$ to obtain a shared seed $a_{i,j}$. The seeds are used to generate individual

and pairwise masks using a pseudorandom generator (PRG). Each client then creates $t$-out-of-$n$ Shamir shares (SS) of $sk_i$ and $b_i$, and sends one share of each to every other client.

In the second round, each client uses the masks generated in round one to compute masked updates according to Eq. 1, which are then sent to the server. All clients perform the ZKP authentication protocol described in Sec. 2 on their update. This ensures that clients use consistent update values across different steps (item ② in threat model). In addition, each client proves, in zero-knowledge, that their sent value $v_i$ is correctly computed as shown in Alg. 2. Specifically, the circuit that is evaluated in zero-knowledge expands the generated seeds to masks, and computes the masked update using Eq. 1. The value of $check$ is then opened by the client, and the server verifies that $check = 1$. This ensures that the masks are correctly generated from seeds, and the masked update is correctly computed (item ③ in the threat model). Users that fail to provide the proof are dropped in the next round and their update is not incorporated in aggregation.

We introduce optimizations in Sec 3.3 that allow the server to derive a bound $q$, for the number of model updates to be checked, such that the probability of detecting Byzantine updates is higher than a predefined rate. The server samples $q$ random parameters from client $i$, and performs the update correctness check (Alg. 2). We note that clients are not motivated to modify the seeds for creating masks, since this results in uncontrollable, out-of-bound errors that can be easily detected by the server (item ④ in threat model). We discuss the effect of using wrong seeds in Appendix B.

In round 3, the server performs unmasking by asking for shares of $sk_i$ for dropped users and shares of $b_i$ for surviving users, which are then used to reconstruct the pairwise and individual masks for dropped and surviving users respectively. The server is then able to obtain the aggregate result.

## 3.2  Establishing Robustness

**Deriving Dynamic Bounds.** To identify the malicious gradient updates, we adaptively find the valid range for *acceptable* gradients per iteration. In this context, acceptable gradients are those that do not harm the central model's convergence when included in the gradient aggregation. To successfully identify such gradients, we rely on the underlying assumption that benign model updates are in the majority while harmful Byzantine updates form a minority of outlier values. In the presence of outliers, the median can serve as a reliable baseline for in-distribution values [14].

In the secure FL setup, the true value of the individual updates is not revealed to the server. Calculating the median on the masked updates is therefore nontrivial since it requires sorting the values which incurs extremely high overheads in secure domain. We circumvent this challenge by forming clusters of users, where our secure aggregation can be used to efficiently compute the average of their updates. The secure aggregation protects user's individual updates, but reveals the final mean value for each cluster $\{\boldsymbol{\mu_1}, \boldsymbol{\mu_2}, ..., \boldsymbol{\mu_c}\}$ to the server. The server can thus easily compute the median ($\boldsymbol{\lambda}$) on the mean of clusters in plaintext.

Using the Central Limit Theorem for Sums, cluster means follow a normal distribution $\boldsymbol{\mu_i} \sim \mathcal{N}(\boldsymbol{\mu}, \frac{1}{\sqrt{n_c}}\boldsymbol{\sigma})$ where $\boldsymbol{\mu}$ and $\boldsymbol{\sigma}$ denote the mean and standard deviation of the original model updates and $n_c$ is the cluster size. We can thus use the standard deviation of the cluster means ($\boldsymbol{\sigma_\mu}$) as a distance metric for marking outlier updates. The distance is measured from the median of means $\boldsymbol{\lambda}$, which serves as an acceptable model update drawn from $\mathcal{N}(\boldsymbol{\mu}, \frac{1}{\sqrt{n_c}}\boldsymbol{\sigma})$. For a given update $\boldsymbol{u_i}$, we investigate Byzantine behavior by checking $|\boldsymbol{u_i} - \boldsymbol{\lambda}| < \boldsymbol{\theta}$, where $\boldsymbol{\theta} = \eta.\boldsymbol{\sigma_\mu} = \frac{\eta}{\sqrt{n_c}}\boldsymbol{\sigma}$. The value of $\eta$ can be tuned based on cluster size ($n_c$) and the desired statistical bounds on the distance in terms of the standard deviation of model updates ($\boldsymbol{\sigma}$). Specifically, assuming a higher bound on the portion of malicious users $\phi_{max}$, the server can automatically adjust $\eta$ such that at most $(1 - \phi_{max}) \cdot n$ of the users are marked as benign where $n$ is the total user count.

**Secure Robustness Check.** We use ZKPs to identify malicious clients that send invalid updates, without compromising clients' privacy. Our ZKP relies on the robustness metrics derived in Sec. 3.2, i.e., the median of cluster means $\boldsymbol{\lambda}$ and the threshold $\boldsymbol{\theta}$. Clients ($\mathcal{P}$) prove to the server ($\mathcal{V}$) that their updates comply with the robustness range check. During the aggregation round in step 1, clients authenticate their private updates, and the authenticated value is used in steps 2 and 3. This ensures that consistent values are used across steps and clients can not change their update after learning $\boldsymbol{\lambda}$ and $\boldsymbol{\theta}$ to fit in the robustness threshold. In step 2, the server makes $\boldsymbol{\lambda}$ and $\boldsymbol{\theta}$ public. Inside ZKP, the clients' updates $\boldsymbol{u_i}$ are used in a Boolean circuit determining if $|\boldsymbol{u_i} - \boldsymbol{\lambda}| < \boldsymbol{\theta}$ as outlined in Alg. 3. Invalid model updates that fail the range check are dropped from the final aggregation round.

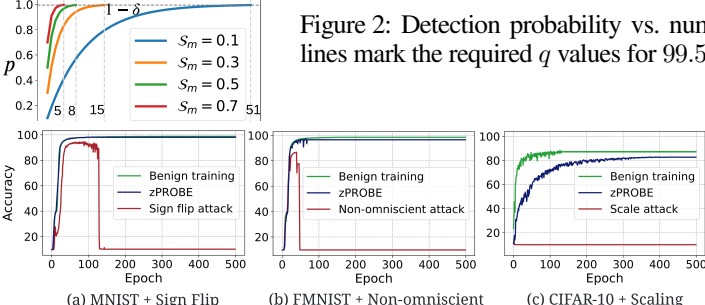

Figure 2: Detection probability vs. number of ZKP checks ($q$). Vertical lines mark the required $q$ values for 99.5% detection rate.

| (a) MNIST + Sign Flip | (b) FMNIST + Non-omniscient | (c) CIFAR-10 + Scaling |
|---|---|---|

| Aggregator | IID | non-IID |
|---|---|---|
| AVG | $93.2 \pm 0.2$ | $92.7 \pm 0.3$ |
| KRUM | $91.6 \pm 0.3$ | $53.1 \pm 3.9$ |
| CM | $91.9 \pm 0.2$ | $78.6 \pm 3.1$ |
| CClip | $93.0 \pm 0.2$ | $91.2 \pm 0.5$ |
| RFA | $93.2 \pm 0.2$ | $92.6 \pm 0.2$ |
| zPROBE | $99.0 \pm 0.0$ | $98.6 \pm 0.4$ |

Figure 3: Test accuracy during benign training (green), Byzantine training without defense (maroon), and Byzantine training with zPROBE.

Table 1: Accuracy of zPROBE and prior robust FL aggregators for non-IID data across 10 runs.

## 3.3 Probabilistic Optimizations

This section provides statistical bounds on the number of required checks to accurately detect malicious clients. Using the derived bounds, we optimize our framework for minimum overhead, thereby ensuring scalability to large models. Malicious clients can compromise their update, by sending updates with distance margins larger than the tolerable threshold $\theta$, or sending incorrect masked updates (Eq. 1). Assume that a portion of model updates $\mathcal{S}_m$, are compromised. The probability of detecting a malicious update is equivalent to finding at least one compromised parameter gradient:

$$p = 1 - \binom{l \cdot (1 - \mathcal{S}_m)}{q} \Big/ \binom{l}{q}, \tag{2}$$

where $l$ is the total model parameter updates, and $q$ denotes the number of per-user ZKP checks on model updates. The above formulation confirms that it is indeed not necessary to perform ZKP checks on all parameter updates within the model. Rather, $q$ can be easily computed via Eq. 2, such that the probability of detecting a compromised update is higher than a predefined rate: $p > 1 - \delta$. Fig. 2 shows the probability of detecting malicious users versus number of ZKP checks for a model with $l = 60K$ parameters. As seen, zPROBE guarantees a failure rate lower than $\delta = 0.005$ with very few ZKP checks. Note that malicious users are incentivized to attack a high portion of updates to increase their effect on the aggregated model's accuracy. We leverage Eq. 2 to derive the required number of correctness and robustness checks as described in Alg. 2 and Alg. 3. For each check, the server computes the bound $q$, then samples $q$ random indices from model parameters for each client. The clients then provide ZKPs for the selected set of parameter indices.

## 4 Experiments

Details of the benchmarked models, datasets, and defense implementation are provided in Appendix C.

**Byzantine Attacks.** We assume 25% of the clients are Byzantine, which is a common assumption in the literature [10]. Malicious users alter a portion $\mathcal{S}_m$ of benign model updates according to a Byzantine attack scenario. We evaluate three commonly observed Byzantine attacks: *Sign Flip* [8], *Scaling* [9], and *Non-omniscient attack* [10]. Attack details are included in Appendix D.

**Baseline Defenses.** We present comparisons with prior work on robust and private FL, i.e., BREA [21] and EIFFeL [22]. While zPROBE is able to implement popular defenses based on rank-based statistics, EIFFeL is limited to static thresholds and requires access to clean public datasets. BREA implements multi-Krum [23], but leaks pairwise distances of clients to the server. zPROBE achieves lower computation complexity compared to both works and higher accuracy[2] compared to EIFFeL. We also benchmark a commonly used aggregator which uses only the median of cluster means, and show that its accuracy is drastically lower that zPROBE. Additionally, we evaluate zPROBE when the training data is non-IID and show our adaptive bounds outperform the state-of-the-art defense [24][3].

## 4.1 Defense Performance

**IID Training Data.** We evaluate zPROBE on various benchmarks using $n = 50$ clients, randomly grouped into $c = 7$ clusters. In Section 4.3 we present evaluations with different number of clients between 30 and 200. Consistent with prior work [10], we assume malicious users compromise all model updates to maximize the

---

[2]Raw accuracy numbers are not reported for BREA, therefore, direct comparison is not possible.

[3]Note that this work focuses on plaintext robust training and does not provide secure aggregation.

accuracy degradation. Fig. 3 demonstrates the convergence behavior of the FL scheme in the presence of Byzantine users with and without `zPROBE` defense. As seen, `zPROBE` successfully eliminates the effect of malicious model updates and recovers the ground-truth accuracy. We show evaluations of `zPROBE` accuracy on other variants of the dataset and attack in Fig. 7 in Appendix E. On the MNIST benchmark, the byzantine attacks cause the central model's accuracy to reduce to nearly random guess (10.2%-11.2%), without any defense. `zPROBE` successfully thwarts the malicious updates, recovering benign accuracy within 0.0%-0.6% margin. On F-MNIST, we recover the original $\sim 88\%$ drop of accuracy caused by the attacks to 0%-2% drop. Finally, on CIFAR-10, the gap between benign training and the attacked model is reduced from 45%- 90% to only 3%-7%. Compared to EIFFeL [22], `zPROBE` achieves 1.2%, 0.5%, and 2.8% higher accuracy when evaluated on the same attack applied to MNIST, FMNIST, and CIFAR-10, respectively.

**Non-IID Training Data.** Most recently, [24] show user clustering over existing robust aggregation methods can adapt them to heterogeneous (non-IID) data. We follow their training setup and hyperparameters to distribute the MNIST dataset unevenly across 25 users. As shown in Tab. 1, `zPROBE` defense outperforms the accuracies obtained by the various defenses evaluated in [24]. This performance boost we believe can be attributed to 1) the use of rank-based statistics to establish dynamic thresholds, and 2) the use of all benign gradients in the aggregation, rather than replacing all values with a robust aggregator, e.g., as in KRUM.

### 4.2 Runtime and Complexity Analysis

Tab. 2 summarizes the total runtime for clients in `zPROBE` for one round of federated training with $n=50$, $c=7$, and $\mathcal{S}_m=0.3$ across different benchmarks. We use the secure aggregation protocol of [6] as our baseline, which does not provide security against malicious clients or robustness against Byzantine attacks. We show the effect of $\mathcal{S}_m$ on `zPROBE` runtime in Tab. 5 in Appendix F. We also show the effect of increasing the number of clients $n$ on `zPROBE` runtime in Tab. 6 in Appendix G.

In Tab. 2, we can see that as the underlying model gets much larger (growing $\sim 4\times$ in size from the MNIST to CIFAR-10 tasks) `zPROBE` overhead grows a negligible amount. This is a strong indicator of the scalability of our proposed secure aggregation. Alongside this, the probabilistic optimizations explained in Sec. 3.3 become more beneficial as the model size increases. Compared to a naive implementation where $1-\mathcal{S}_m$ parameters are checked, we achieve a speedup of 3 orders of magnitude in client and server runtime.

| Dataset | Baseline (ms) | `zPROBE` (ms) |
|---------|---------------|---------------|
| MNIST | 208.038 | 424.399 |
| F-MNIST | 214.371 | 432.659 |
| CIFAR-10 | 231.23 | 440.911 |

Table 2: Runtime of `zPROBE` secure aggregation vs. the baseline protocol of [6] with no support for robustness and malicious clients.

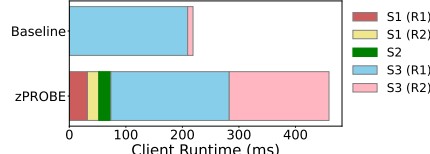

Figure 4: Runtime breakdown for CIFAR-10, corresponding to rounds (R) from Alg. 1 and steps (S) from Fig. 1..

We also provide the detailed breakdown of the runtime for various components of `zPROBE` in Fig. 4. Results are gathered on CIFAR-10 dataset and ResNet-20 architecture with $n=50$, $c=7$, and $\mathcal{S}_m=0.3$. Step 2 has very low overhead, even when only 30% of model updates are Byzantine, which requires more checks. The most significant operation in terms of percent increase from baseline is observed in Step 3 (R2), where the correctness of masked updates are checked (Alg. 1 Round 2 and Alg. 2). `zPROBE` enjoys a low communication overhead as well, requiring only 2.1MB and 4.4MB of client and server communication respectively, for a round of aggregation over CIFAR-10. Overall, with sub-second performance on all benchmarks examined, `zPROBE` provides an efficient full privacy-preserving and robust solution for FL.

**zPROBE Complexity.** In this section we present the complexity analysis of `zPROBE` runtime with respect to number of clients $n$ (with $k=\log n$) and model size $l$.

• Client: Each client computation consists of performing key agreements with $O(k)$, generating pairwise masks with $O(k \cdot l)$, creating t-out-of-k Shamir shares with $O(k^2)$, performing correctness checks of Alg. 2 with $O(k \cdot l)$, and performing robustness checks of Alg. 3 with $O(l)$. The complexity of client compute is therefore $O(\log^2 n + l \cdot \log n)$.

• Server. The server computation consist of reconstructing t-out-of-k shamir shares with $O(n \cdot k^2)$, generating pairwise masks for dropped out clients with $O(n \cdot k \cdot l)$, performing correctness checks of Alg. 2 with $O(n \cdot l)$, and performing robustness checks of Alg. 3 with $O(n \cdot l)$. The overall complexity of server compute is thus $O(n \cdot \log^2 n + n \cdot l \cdot \log n)$.

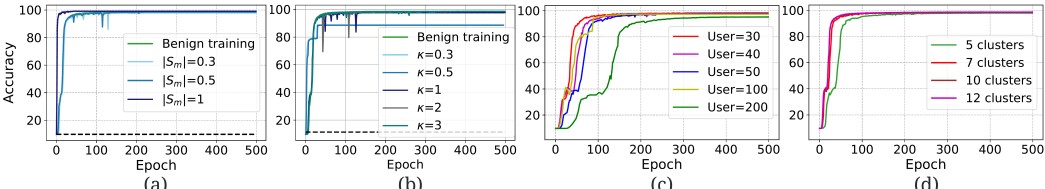

Figure 5: Analysis of `zPROBE` defense performance with varying (a) portion of compromised updates, (b) attack magnitude, (c) number of clients, and (d) number of user clusters. The dashed line in (a), (b) corresponds to the highest test accuracy obtained during training when no defense is applied.

We are unable to directly compare `zPROBE`'s runtime numbers with previous private and robust FL methods since their implementations are not publicly available. Instead, Tab. 3 presents a complexity comparison between `zPROBE`, BREA [21], and EIFFeL [22] with respect to number of clients $n$ (with $k = \log n$), model size $l$, and number of malicious clients $m$. `zPROBE` enjoys a lower computational complexity compared to both prior art for client and server. Specifically, the client runtime is quadratic and linear with number of clients in BREA and EIFFeL respectively, whereas logarithmic in `zPROBE`.

|        | Client          | Server                                          |
|--------|-----------------|-------------------------------------------------|
| BREA   | $O(n^2l+nlk^2)$ | $O((n^3+nl)k^2 \cdot \log(k))$                  |
| EIFFeL | $O(mnl)$        | $O((n+l)nk^2 \cdot \log(k)+m.l.\min(n,m^2))$    |
| zPROBE | $O(k^2+kl)$     | $O(nk^2+nlk)$                                   |

Table 3: Runtime complexity of `zPROBE` vs. prior works BREA [21] and EIFFeL [22].

### 4.3 Discussion

We perform a sensitivity analysis to various attack parameters and FL configurations on F-MNIST. Number of clients is set to $n = 50$ with $c = 7$ clusters, unless otherwise noted. Sign flip attack [8] is applied to all model updates with $\kappa = 5$. As shown, `zPROBE` is largely robust to changes in the underlying attack or training configuration, consistently recovering the central models' accuracy.

**Portion of Compromised Updates $\mathcal{S}_m$.** We vary the portion of Byzantine model updates ($\mathcal{S}_m$) and show the accuracy of the central model with and without `zPROBE` robustness checks in Fig. 5(a). Even when only a small portion of model updates are malicious, `zPROBE`'s outlier detection can successfully recover the accuracy from random guess ($10\%$) to $97.9\%$. To worsen the central model's accuracy, Byzantine workers are incentivized to attack a high number of model updates. Attacking all model updates results in a $89.7\%$ drop of accuracy when no defense is present. However, even when all model updates are compromised, `zPROBE` recovers the central model's accuracy with less than $0.5\%$ error margin.

**Attack Magnitude.** We control the magnitude of the perturbation applied to model updates by changing the parameter $\kappa$ in various Byzantine attack scenarios. Fig. 5(b) shows the effect of the attack magnitude on the central model's accuracy and `zPROBE`'s defense performance. As seen, a higher perturbation is easier to detect using our median-based robustness check. The Byzantine attack can cause an accuracy drop of $\sim 88\%$ when no robustness check is applied. However, `zPROBE` can largely recover the accuracy degradation, reducing the accuracy loss to $0.2\%$-$9.8\%$.

**Aggregation Strategy.** Recall from Fig. 1 that `zPROBE` uses the median of per-cluster means (in plaintext) to extract a threshold, which is then used in step 2 to filter out malicious clients. Rather than performing steps 2 and 3 of `zPROBE`, an alternative robust aggregation rule may directly use the median value to update the global model. Merely using the median for the final aggregation ignores beneficial updates by benign users. This leads to a drastic accuracy degradation of $28.6\%$ compared to `zPROBE` which includes all gradients that pass the threshold. A comparison between median-based aggregation and `zPROBE` is presented in Appendix H, Fig. 8.

**Number of Clients ($n$).** Fig. 5(c) shows the convergence of `zPROBE` during training for various $n \in [30, 200]$. We note that $n$ is the subset of clients which are picked for an aggregation round. We pick this range according to practical deployments of FL where the server picks a small fraction of all FL clients for each aggregation round [25]. As seen, client count does not affect `zPROBE` convergence and the central model's final accuracy. Specifically, `zPROBE` can scale to $n = 200$, which is among the largest studied user counts in robust and private FL.

**Number of Clusters ($c$).** Fig. 5(d) shows the effect of number of clusters on accuracy. The performance of `zPROBE` is largely independent of the number of clusters, showing less than $0.18\%$ variation for different

$c$ while the increase in latency is less than $8\%$. The number of clusters can therefore be selected freely such that user privacy is ensured. Recall from Fig. 1 that cluster means in step 1 of zPROBE are revealed to the server. To analyze the privacy implications, we rely on the contemporary literature in model/gradient inversion which show that increasing the batch size, or equivalently number of users, ($> 100$ [26] or $>48$ [27]) reduces the effectiveness of such attacks.

We benchmark the SOTA attack by [26] to reconstruct user data from the aggregate. We use a small 4-layer model and a batch size of 10 to benefit the attacker. Fig. 10(a),(b) in Appendix J show the efficacy of the inversion attack as the number of users in the aggregation varies. As seen, the reconstructions are unintelligible with $>4$ users per cluster. More recently [28] quantify the user information leakage from the aggregate value using Mutual Information (MI). They show that MI reduces with more clients, starting to plateau around 10-20 users where the reconstructed image quality of the DLG attack [29] is severely affected. Based on these results, our cluster size range of 7 to 30 can preserve user privacy.

**User Dropout.** zPROBE secure aggregation supports user dropouts, i.e., when a user is disconnected amidst training iterations and/or in between zPROBE steps (see Fig. 1). We simulate the effect of user dropout on the training accuracy of zPROBE in Appendix I, Fig. 9. As shown, the changes in the training accuracy are very subtle ($<0.13\%$ difference) in the presence of random user dropouts.

## 5    Related Work

**Secure Aggregation.** Cryptographic techniques have been used in prior work for secure aggregation, e.g., by using random masks to hide private updates [5, 6] during computation. A line of work [30, 31] considers a different trust model with non-colluding servers that receive secret shares of data and collaborate to compute the aggregate. However, realizing the non-colluding trust assumption can be challenging in practice. Differential Privacy (DP) can provide complementary privacy guarantees to cryptographic methods (by protecting information leakage from output), and have been used in conjunction to reduce the required noise. [7] combine threshold homomorphic encryption and differential privacy for private aggregation.

**Robust Aggregation.** Prior work on Byzantine-robust aggregation sanitizes updates using robust statistics or historical information. (Multi-)Krum [32, 23] selects updates with minimum Euclidean distance to their neighbors. Coordinate-wise operations based on median and trimmed mean [11, 13] have also been proposed that calculate the mean of values closest to the median for each coordinate. More recently, AKSEL [14] defines an interval around the median and aggregates values within that interval. [15, 33] use the robustness of geometric median to provide a robust update rule. Bulyan [16] augments prior aggregation rules to ensure all coordinates are agreed upon by a majority of user gradients.

Several works propose applying robust aggregation over an accumulated history of gradients, assuming IID data. [34] use the concentration of aggregated past gradients around the median to mark byzantine workers. Similarly, [35] apply Byzantine-resilient aggregation rules on a weighted average of past gradients using a momentum term. In lieu of using the median, centered clipping [36] iteratively scales the accumulated gradients to ensure robust aggregation. The aforesaid works focus on robust aggregation under IID data assumptions. [24] propose user clustering as an effective way to adapt previously proposed robust aggregation methods, e.g., Krum, to heterogeneous (non-IID) data. zPROBE designs a *new* aggregation rule that 1) enables efficient execution in the secure domain and 2) achieves state-of-the-art accuracy compared to [24].

**Robust and Secure Aggregation.** RoFL [37] focuses on model poisoning, when malicious users try to embed a backdoor in the model, without downgrading accuracy on benign data. RoFL uses Pedersen commitments to implement ZKP of norm bounds over model updates. RoFL does not support dropouts during aggregation and the proposed $l$-norm bounds are unsuitable against Byzantine workers. zPROBE considers Byzantine attacks where the malicious parties send invalid updates to degrade the central model's accuracy. In this domain, BREA [21] relies on Shamir secret sharing and multi-Krum [23] for aggregation. However, BREA reveals the pairwise distances of client updates to the server, i.e., even one client's collusion with the server would reveal all updates.

SHARE [38] incorporates secure averaging [5] on randomly clustered clients, and filters cluster averages through robust aggregation. Any cluster with malicious clients detected will be dropped, resulting in loss of all benign updates. SHARE's mitigation is to repeat the random clustering several times for each epoch, resulting in increased computation and communication cost. Moreover, reclustering compromises privacy as the server observes different variations of cluster averages which can leak information about the user updates. Most recently, EIFFeL [22] proposes a robust aggregation using Shamir shares of client updates, and secret-shared non-interactive proofs (SNIP). EIFFeL does not support rank-based statistics for robustness checks, resulting in higher accuracy degradation, and requires access to a clean public dataset for defense parameters.

## 6 Conclusion

This paper presented the `zPROBE` framework for low overhead, scalable, private, and robust FL in the malicious client setting. `zPROBE` ensures correct behavior from clients, and performs robustness checks on model updates, combining zero-knowledge proofs and secret sharing to provide robustness *and* privacy guarantees. `zPROBE` presents a paradigm shift from previous work by providing a private robustness defense relying on rank-based statistics with cost that grows sublinearly with respect to number of clients.

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

## A  Preliminary on ZKP

zPROBE incorporates the Wolverine ZKP protocol [17]. In Wolverine, value $x$ known by $\mathcal{P}$ is authenticated using information-theoretic message authentication codes (IT-MACs) [39] as follows: assume $\Delta$ is a global key sampled uniformly and is known only to $\mathcal{V}$. $\mathcal{V}$ is given a uniform key $K[x]$ and $\mathcal{P}$ is given the corresponding MAC tag $M[x] = K[x] + \Delta.x$. An authenticated value can be *opened* (verified) by $\mathcal{P}$ sending $x$ and $M[x]$ to $\mathcal{V}$ to check whether $M[x] \overset{?}{=} K[x] + \Delta.x$.

Wolverine consists of an interactive offline phase (which is circuit and input independent), and an online phase that builds on a special case of a secure two-party computation where one party has no input. The parties prepare authenticated values for $\mathcal{P}$ inputs, and the computation is represented as an arithmetic or Boolean circuit which is evaluated gate-by-gate. All communication during the online phase is from the $\mathcal{P}$ to the $\mathcal{V}$, therefore the circuit can be evaluated using only one round of communication. At the end of the evaluation, $\mathcal{P}$ reveals the output which indicates the correctness of the proof.

## B  Malicious Seed Modification

Fig. 6(a) demonstrates the histogram of $L_\infty$ norm of *benign* gradients, observed throughout training across 100 users for CIFAR-10 dataset. As seen, majority of the gradient norms are bounded in $[0, 0.25]$. Masks are generated using seeds through a pseudorandom generator (PRG), and malicious users cannot control the resulting error when changing the seed. Fig. 6(b) shows a histogram of mask values generated from random seeds over 10000 runs. As shown, changing the seed may cause unpredictable and drastic changes in the mask. By changing the random seed, the generated masks can vary anywhere between $-3 \times 10^4$ to $3 \times 10^4$, which is much larger than the normal observed range for model updates. As such, when a malicious user changes the random seed from which the masks are generated, it can lead to easily recognizable errors in the gradient that raises alarms for the server. Thus, in our threat model malicious users are incentivized to use the correct seed when computing masks.

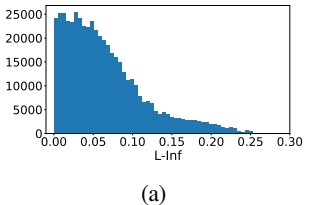
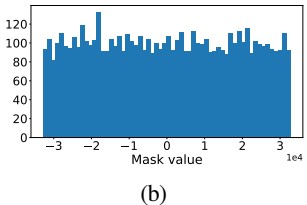

(a)                                    (b)

Figure 6: Histogram of (a) ResNet-20 gradient norms observed during training on CIFAR-10, and (b) mask values when changing the random seed.

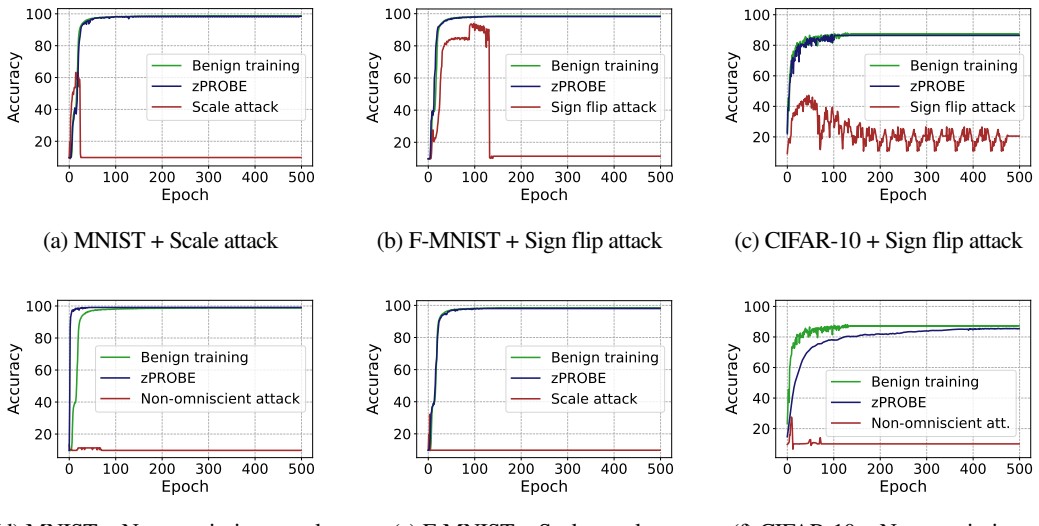

(a) MNIST + Scale attack     (b) F-MNIST + Sign flip attack     (c) CIFAR-10 + Sign flip attack

(d) MNIST + Non-omniscient attack     (e) F-MNIST + Scale attack     (f) CIFAR-10 + Non-omniscient

Figure 7: Test accuracy as a function of FL training epochs for different attacks and benchmarks. Each plot shows the benign training (green), Byzantine training without defense (maroon), and Byzantine training in the presence of zPROBE defense.

## C    Experimental Setup Details

**Dataset and Models.** We consider three benchmarks commonly studied by prior work in secure FL. Our first benchmark is a variant of LeNet5 [40] trained on the MNIST dataset [41], with 2 convolution and 3 fully-connected layers, totaling $42K$ parameters. Our second benchmark is the Fashion-MNIST (F-MNIST) dataset [42] trained on the LeNet5 architecture with $60K$ parameters. Finally, to showcase the scalability of our approach, we evaluate ResNet-20 [43] with $273K$ parameters trained on the CIFAR-10 dataset [44] which is among the biggest benchmarks studied in the secure FL literature [37, 22]. Table 4 encloses the training hyperparameters for all models.

Table 4: Training hyperparameters.

| Benchmark | # Clients | LR | # Epochs | Batch size (per user) |
|---|---|---|---|---|
| MNIST (IID) + LeNet5 | 50 | 0.01 | 500 | 12800 (256) |
| MNIST (non-IID) + LeNet5 | 25 | 0.01 | 500 | 800 (32) |
| F-MNIST (IID) + LeNet5 | 50 | 0.01 | 500 | 12800 (256)[†] |
| CIFAR-10 (IID) + ResNet-20 | 50 | 0.05 | 500 | 12800 (256) |

[†]When varying the number of clients, we keep the total batch size as 12800 and scale the per user batch size accordingly.

**Implementation and Configuration.** zPROBE defense is implemented in Python and integrated in PyTorch to enable model training. We use the EMP-Toolkit [45] for implementation of zero-knowledge proofs. We run all experiments on a 128GB RAM, AMD Ryzen 3990X CPU desktop. All reported runtimes are averaged over 100 trials.

## D Byzantine Attacks

We show the effectiveness of our robustness checks against three commonly used Byzantine attacks. Here $\mathcal{U}_m$ denotes the malicious updates, where $|\mathcal{U}_m| = \mathcal{S}_m \cdot l$ and $l$ is the total number of model updates.

- *Sign Flip* [8]. Malicious client flips the sign of the update: $u = -\kappa.u, \kappa > 0 \ (\forall u \in \mathcal{U}_m)$
- *Scaling* [9]. Malicious client scales the local gradients to increase the influence on the global model: $u = \kappa.u, \kappa > 0 \ (\forall u \in \mathcal{U}_m)$
- *Non-omniscient attack* [10]. Malicious clients construct their Byzantine update by adding a scaled Gaussian noise to their original update with mean $\mu$ and standard deviation $\sigma$: $u = \mu - \kappa.\sigma \ (\forall u \in \mathcal{U}_m)$

## E `zPROBE` Test Accuracy

Fig. 7 shows the test accuracy of `zPROBE` in face of different variations of Byzantine attacks and datasets. The dataset is distributed evenly (IID) among $n = 50$ clients. The server randomly clusters users into $c = 7$ groups during each training round. We assume malicious users compromise all model updates $|\mathcal{S}_m| = 1$ to maximize the accuracy degradation.

## F Effect of $\mathcal{S}_m$ on `zPROBE` Runtime

Tab. 5 summarizes the runtime of `zPROBE` versus the portion of attacked model updates. By decreasing $\mathcal{S}_m$, `zPROBE` requires more checks to detect the outlier gradients as outlined in Eq. 2. Nevertheless, due to the optimizations in `zPROBE` robustness and correctness checks, we are still able to maintain sub-second runtime and sublinear growth with respect to number of ZKP checks necessary.

Table 5: `zPROBE` performance for LeNet5 on F-MNIST vs. the portion of Byzantine model updates ($\mathcal{S}_m$).

|  | $\mathcal{S}_m$ | | | | |
|---|---|---|---|---|---|
|  | 0.1 | 0.3 | 0.5 | 0.7 | 1.0 |
| # ZKP Checks | 51 | 15 | 8 | 5 | 1 |
| `zPROBE` Runtime (ms) | 777.9 | 452.9 | 372.6 | 349.6 | 316.5 |

## G Effect of Number of Clients on `zPROBE` Runtime

Fig 6 shows the effect of increasing the number of clients on the performance on `zPROBE`. For these experiments, results are gathered on the F-MNIST dataset, with $c = 7$ and $|\mathcal{S}_m| = 0.3$. As seen, although exceeding the sub-second performance as the number of clients scales up, `zPROBE` maintains sublinear growth in runtime with respect to number of clients.

| # Clients | `zPROBE` Runtime (ms) |
|---|---|
| 30 | 298.4 |
| 40 | 369.7 |
| 50 | 452.9 |
| 70 | 598.8 |
| 100 | 828.6 |
| 200 | 1620.4 |

Table 6: Runtime of `zPROBE` over varying number of clients

## H Effect of Aggregation Method on Accuracy

`zPROBE` leverages the median of averaged model updates across user clusters to check whether the incoming updates are benign or Byzantine. An alternative aggregation strategy is to directly use the median of cluster means, rather than performing the subsequent per-user checks. Fig. 8 shows the test accuracy of `zPROBE` as training progresses, when compared to the above baseline aggregation method that applies

the coordinate-wise median of cluster means. As seen, compared to `zPROBE`, this baseline suffers from a large accuracy degradation, since all information in benign user updates is lost by replacing the aggregation with the median, which can potentially contain Byzantine workers.

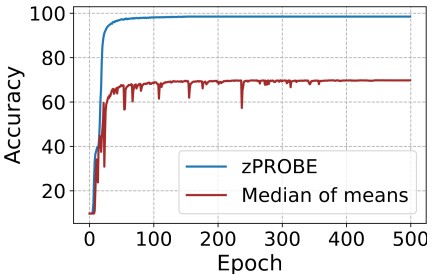

Figure 8: Test accuracy of `zPROBE` compared with an aggregation methodology that uses the median of cluster means to update the model.

## I  Effect of User Dropout

Fig 9 shows the effect of random user dropouts on `zPROBE` defense. As shown, the fluctuations in the central model's test accuracy are negligible ($<0.13\%$). The robustness of `zPROBE` aggregation protocol to user dropouts is intuitive since the remaining users can carry on the training.

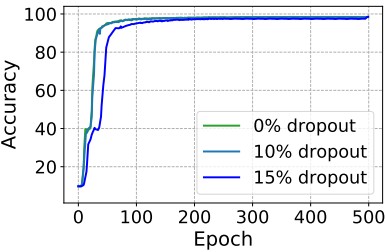

Figure 9: The effect of (a) number of clusters, and (b) user dropout on defense.

## J  Effect of Cluster Size on Inversion Attack

Fig. 10 shows the effect of cluster size on gradient inversion attacks. In Fig 10(a) we show the effectiveness of the attack [26] for different cluster sizes. Fig 10(b) represents the reconstruction results from user data for different number of users participating in the aggregation round.

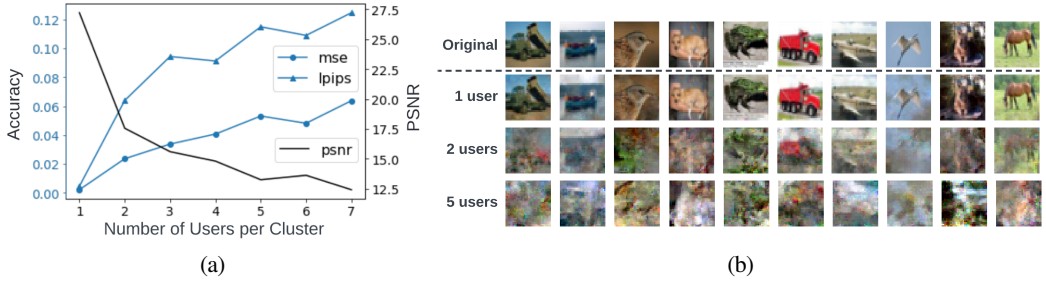

(a)                                                          (b)

Figure 10: Performance of gradient inversion attacks for different cluster sizes.

