# OpenReview forum: "zPROBE: Zero Peek Robustness Checks for Federated Learning"
_NeurIPS.cc/2022/Workshop/TSRML — TSRML2022_

### Official Review · Reviewer_jQBY · 2022-10-20
**Good paper, accept**

**Overall Recommendation:** Overall I think this paper is a good …
**Overall Rating:** 7

**Summary:**

This paper investigates the problem of combatting with Byzantine attacks under the privacy-preserving federated learning setting. It removes updates that are too far away from the median that are potentially malicious updates. The median is hard to compute under the privacy-preserving setting, so the paper proposes to first cluster the clients into several groups, and then compute the median of the means of these groups. It demonstrates the effectiveness of the proposed algorithm on real datasets with experiments.

**Strengths:**

Significance: The paper addresses the problem of loss of security when preserving privacy in federated learning, which is an interesting and important problem.

Quality: The proposed method is very intuitive and makes a lot of sense. The experiments show that it can achieve good performances and is more efficient than previous methods.

Clarity: The paper is well written and easy to follow.

**Weaknesses:**

Novelty: I think the proposed method is a combination of a number of existing methods instead of a completely new one. The part that is novel is computing the median by first clustering the clients. Nevertheless, I still think the overall implementation is good, and is more efficient than previous methods.

Format: The paper is longer than the soft page limit (6 pages), with many space adjustments.

**Review Confidence:**

3: The reviewer is fairly confident that the evaluation is correct

---

### Official Review · Reviewer_tJ6D · 2022-10-23

**Overall Rating:** 7

**Summary:**

In this work, the authors provide a secure aggregation method for federated learning to defend against byzantine attacks. Specifically, the authors utilizes random clustering and asks clients to provide a ZKP check that the model updates is close to the median. Empirical results have shown the proposed method is able to outperform prior baselines on different byzantine attacks.

**Strengths:**

This work provides a novel method for robust federated learning. Empirical results are very promising. The paper is also well-written and well-organized.

**Weaknesses:**

- The work would be strengthened if any formal robustness analysis is provided. It is not clear whether sign flip is the strongest attack in this case. Therefore, any certificate for the proposed method would make the paper stronger.
- Related to the previous point, could the authors provide experimental results against label flipping attacks (where one randomly flip the label of the true training data) + scaling?
- Apparently, the length of this paper is 8 pages, which is against the workshop's instruction of maximum 6 pages. I would strongly recommend the authors to shorten the paper within the correct page limits if the paper got accepted to the workshop.

**Overall Recommendation:**

Interesting paper with strong empirical results. However, page length does not satisfy the workshop requirement. Still recommend the paper to be presented at the workshop.

**Review Confidence:**

3: The reviewer is fairly confident that the evaluation is correct

---

### Decision · Program_Chairs · 2022-10-23

**Decision:**

Accept

**Comment:**

Following the unanimous recommendations from reviewers, the submission is accepted.